# Association between malaria and undernutrition among pregnant women at presentation for antenatal care in health facilities in the Mount Cameroon region

Vanessa Tita Jugha[1]*, Juliana Adjem Anchang[2], Germain Sotoing Taiwe[1], Helen Kuokuo Kimbi[1,3,4], Judith Kuoh Anchang-Kimbi[1]

1 Department of Animal Biology and Conservation, University of Buea, Buea, Cameroon, 2 International Centre for Agricultural Research in the Dry Areas, ICARDA, Cairo, Eygpt, 3 Department of Biomedical Sciences, University of Bamenda, Bamenda, Cameroon, 4 Department of Microbiology and Immunology, Drexel University College of Medicine, Philadelphia, Pennsylvania, United States of America

* jughav@yahoo.com

**Data Availability Statement:** All relevant data are within the paper and its Supporting Information files

## Abstract

In resource limited settings, malaria and undernutrition are major public health problems in pregnancy. Therefore, this study assessed the association between malaria infection and undernutrition among pregnant women in the Mount Cameroon area. This cross-sectional study enrolled 1,014 pregnant women consecutively over a year. A structured questionnaire was used to collect socio-demographic information and clinical data. Maternal nutrition was assessed using dietary diversity (DD). Peripheral blood samples collected were used for the diagnosis of malaria parasitaemia by microscopy whereas haemoglobin (Hb) levels were determined using an Hb meter. Logistic regression was used to determine factors associated with malaria and dietary diversity. The prevalence of malaria infection and undernutrition was 17.8% and 89.6% respectively. In addition, of those infected with malaria, geometric mean parasite density was 301/µL of blood (range: 40–9280) while mean DD score was 3.57±0.82 (range: 1–7). The odds of being infected with malaria parasitaemia was highest among women enrolled in the rainy season (OR = 1.58, P = 0.043), who were farmers (OR = 2.3, P = 0.030), had a household size of < 4 individuals (OR = 1.48, P = 0.026) and who were febrile (OR = 1.87, P < 0.001). Also, attending clinic visits in Mutengene Medical Centre (OR = 2.0, P = 0.012) or Buea Integrated Health Centre (OR = 2.9, P = < 0.001), being < 25 years (OR = 2.4, P = 0.002) and a farmer (OR = 10.6, P = 0.024) as well as < 4 clinic visits (OR = 1.62, P = 0.039) were identified as predictors of undernutrition. Furthermore, the association between malaria and DD was statistically significant (P = 0.015). In this study, undernutrition was highly frequent than malaria infection. Thus, there is an urgent need to improve maternal awareness through nutritional counselling and health campaigns on the benefits of consuming at least five food groups. Besides, improved maternal dietary nutrient intake is likely to have impact on the burden of malaria parasite infection.

**Funding:** The author(s) received no specific funding for this work.

**Competing interests:** The authors have declared that no competing interests exist.

## Introduction

Malaria is a life-threatening parasitic infection disproportionately affecting the poor, children below 5 years of age and pregnant women. In 2020, an estimated 241 million cases of malaria and 627,000 deaths occurred globally, 95% of the cases were documented in the World Health Organization (WHO) African region where majority (99%) of the infection is caused by *Plasmodium falciparum* [1,2]. Furthermore, of the estimated 33.8 million pregnancies that occur in sub-Saharan Africa an area characterized by moderate to high transmission, 11.6 million were exposed to malaria infection [2]. Although infections during pregnancy are frequently asymptomatic and often go undetected, they are however associated with anaemia, intrauterine growth restriction, low birth weight (LBW), stillbirth and maternal death [3,4]. Apart from transmission setting, other factors previously documented to increase the risk of malaria among pregnant women are; age, gravidity, season, level of education, coinfection and nutritional status [5,6]. In order to control and mitigate the adverse effects of malaria in pregnancy (MiP), the WHO recommends a package of interventions which include; use of insecticide-treated nets (ITNs), administration of intermittent preventive treatment in pregnancy with sulphadoxine-pyrimethamine (IPTp-SP) and prompt as well as effective case management [7,8]. In Cameroon, these interventions are usually delivered in antenatal clinics (ANC). Therefore, MiP is a useful marker for malaria surveillance as recommended by the WHO as this helps the country to evaluate the impact of malaria control programs as well as to revise or reinforce already existing intervention strategies.

Malnutrition, on the other hand causes significant morbidity and mortality across a range of health systems and is associated with economic burden [9]. Globally, over 170 million people are undernourished among which 1 billion suffer from micronutrient deficiencies [10,11]. In Africa, the burden of undernutrition due mainly to inadequate dietary diversification is still high with an estimated prevalence of 23.5% [12,13] and pregnant women are regarded as most vulnerable [14,15]. Furthermore, the consequences of inadequate maternal nutrient intake are severe for the new-borns [11] as every year nearly 20 million infants are born with LBW, an early form of malnutrition that is closely linked to maternal nutritional status during pregnancy [11]. Also, poor nutrition during pregnancy increases susceptibility to infections [16,17]. In low-income settings, maternal undernutrition is often measured by anthropometry because it is simple and feasible in clinical settings [18,19]. However, the minimum dietary diversity for women (MDD-W) scale is currently considered the main nutrition indicator in the Sustainable Development Goal (SDG) [20–22]. Although Cameroon is bestowed with a rich agricultural biodiversity, it is not exempted from the burden of undernutrition [23]. Cognizant of this, efforts have been made over the years to raise the nutrition profile to a high priority in the country [24,25]. Despite these positive developments, maternal dietary diversity is still a neglected health problem in Cameroon [26] as much of the evidence pertaining to the interventions focuses on the health benefit of the child. Even though it has been shown that factors influencing malaria might also influence dietary nutrient uptake [27–29], reports on the predictors of maternal dietary diversity in the study area are limited.

In low income settings, malaria and undernutrition frequently coexist in pregnant women and the two may interact to worsen pregnancy outcomes [30]. Evidence in the Democratic Republic of Congo and Kenya indicates that the effect of malaria on foetal growth and birthweight are largely dependent on maternal nutritional status [30–32]. In the context of attaining the SDG of ending all forms of malnutrition and malaria by 2030 [33], interventions to protect pregnant Cameroonian women and their foetuses from poor nutrition and malaria remains poorly integrated as little efforts have been made to comprehensively explore this relationship. Moreover, investigating the association between these ailments will be essential to develop

public health interventions especially in our setting where these two conditions co-exist. Hence, this study sought to examine the association between infection with malaria parasitaemia and poor nutrition among pregnant women in the Mount Cameroon region.

## Methodology

### Study setting

This study was conducted in various health facilities located in the Tiko and Buea Health Districts of the Mount Cameroon region. The Tiko Health District (THD) is situated 18–240 meters above sea level with temperatures ranging between 25.08 to 33°C [26,34]. Moreover, this district hosts the Cameroon Development Corporation (CDC) which is the largest agro-industrial complex in Tiko that grows banana, semi-finished rubber, and palm oil for export. Also, the presence of the local seaport allows for fishing and the exchange of goods between neighbouring countries. In this health area, pregnant women were enrolled from Tiko Holforth Health Centre (THHC) and Mutengene Medical Centre (MMC).

The Buea Health District (BHD) is situated 896 metres above sea level with an average temperature range of 18–27°C. The principal occupational activity here is teaching and business. Besides, study respondents here were enrolled from the Buea Integrated Health Centre (BIHC) and the Mount Mary Hospital (MMH). All the above-mentioned health facilities were chosen because they are highly accessible thus facilitating utilisation of ANC services [3].

The Mount Cameroon region has an equatorial climate characterised by a temperature range of 18–35°C [35,36] and two seasons. The short dry season begins late in October and ends in February while the long rainy season which is usually accompanied by high precipitation (2,000–10,000 mm) [37], spans from March to October. Malaria transmission in the area is perennial with peak periods of infection corresponding to the rainy season [3]. Also, pregnant women residing in the study area are frequently exposed to *P. falciparum* which is the main malaria parasite species in the region with prevalence rates between 13.4% and 22.4% [3,38].

### Study design and population

This cross-sectional study enrolled consenting pregnant women aged 15–49 years in any gestational age for over a year [26]. The sample size for this survey has previously been estimated based on the prevalence of anaemia in the study area [26]. This is because according to the WHO, maternal anaemia in the Mt Cameroon region is still a severe health problem [39]. In brief, it was calculated using the formula $n = Z^2 p(1-p)/d^2$ [40] where n is the required sample size; Z = 1.96 at a 95% confidence level; prevalence (p) = 40% and d, the acceptable error willing to be committed was 0.05. Thus, the minimum sample size was 408 study participants per health district giving a total of 816. After taking into consideration the loss of biological material at 10%, the final sample size was increased to 1,014 participants. Sampling was done using the non-probability sampling technique for convenience and only women who gave their consent were enrolled into the study consecutively.

### Ethics statement

Ethical clearance (Ref No: 2019/967-05/UB/SG/IRB/FHS) for this study was obtained from the Institutional Review Board hosted by the Faculty of Health Sciences, University of Buea and Administrative authorization from the South West Regional Delegation of Public Health, Buea. After explaining study protocol, informed consent forms stating the purpose and benefits of the study as well as the amount of blood to be drawn from each participant was read,

described and distributed. Participation in the study was voluntary. Subsequently, written and signed informed consent was obtained from each individual before enrolment into the study. Pregnant women with evidence of chronic disease and complicated pregnancy (hypertension, pre-eclampsia, diabetes) were not eligible to take part in the study.

### Data acquisition

A pre-tested structured questionnaire through a face-to-face interview conducted by well-trained graduate field assistants was used to document maternal demographic information, education level, marital status, occupation, ITN usage, IFA (iron folic acid) uptake, household size, house type, house ownership, type of toilet, possession/availability of household assets (television, radio, mobile phone, motorcycle, bicycle, car) and source of drinking water. Antenatal characteristics (parity, gestational age, gravidity, trimester of pregnancy, number of clinic visits attended and IPTp-SP dose uptake) and history of fever in the last 48 hours prior to survey were also documented and verified by checking the antenatal card.

### Clinical assessment

Each participants' axillary temperature was measured using a digital thermometer and fever (febrile status) was defined as temperature greater than 37.5˚C [3]. In addition, maternal gestational age (weeks) was determined upon physical examination by the midwife who used either a gestational calendar or ultrasound when the date of last menstrual period was not known.

### Assessment of minimum dietary diversity

Maternal dietary nutrient intake was assessed using the minimum dietary diversity for women (MDD-W) questionnaire endorsed by the Food and Agricultural Organization of the United Nations [21]. Ten food groups provided by the tool developers was used to calculate the dietary diversity (DD) score [21,26]. These food groups were: starchy staples (grains, white roots, tubers and plantains); pulses (beans, peas and lentils); nuts and seeds; dairy products; meat, fish and poultry; eggs; dark green leafy vegetables; vitamin A-rich vegetables and fruits; other vegetables and other fruits [21]. Responses were scores as "1" for women who consumed food item(s) within any food group [21,26]. Scores were then summed up to obtain each participants DD score which was further dichotomized into adequate (good) DD, for pregnant women who consumed at least 5 or more food groups and inadequate (poor) DD, for participants who consumed < 5 food groups.

### Household economic status determination

In this study, maternal household wealth index based on household assets and characteristics was determined using principal component analysis (PCA). Indicators subjected to PCA were; house type (brick or plank/wood), house ownership (yes/no), toilet type (pipped sewer system or pit toilet), possession of household goods (television, radio, mobile phone, car, motorcycle), and source of drinking water (pipped to residence/mineral or public tap/spring). The factor score generated by this analysis was used to categorise each study subject as having either a low or high wealth index.

### Laboratory methods

Two to Three millilitres (ml) of venous blood was collected from each expectant mother using a sterile disposable syringe by a trained laboratory technician. Thick and thin blood films were prepared on the same slide, allowed to air-dry, fixed with absolute methanol for 30 seconds

(thin blood film only) and stained with 10% Giemsa for 15 minutes. Each slide was then examined by two independent microscopists under oil immersion with the 100x objective of a light microscope for the identification of malaria parasites. A slide was declared negative if no parasites were found after examining 100 high power fields. With each positive slide, parasite density/μL of blood was determined by counting the number of parasites per 200 leucocytes on thick blood film after assuming a white blood cell (WBC) count of 8000 leucocytes/μL of blood [41,42]. Symptomatic infection was defined as fever associated with malaria parasites while asymptomatic infection was defined as the presence of malaria parasites in the absence of fever [3]. Parasitaemia levels was categorised as low ($\leq$ 500 parasites/μL of blood), moderate (501–5000 parasites/μL of blood) and high ($\geq$ 5000 parasites/μL of blood) [43,44].

Haemoglobin (Hb) concerntration was determined using Urit12® Hb meter (URIT Medical Electronics Co., Ltd. Guangxi, China). Maternal anaemia status in this study was gestational age specific and defined as follows; Hb < 11.0 g/dL for women between 1–13 weeks and $\geq$ 27 weeks of gestation and Hb < 10.5 g/dL for women between 14–26 weeks of gestation [45]. This cut-off used takes into consideration the haemodilution effect of pregnancy which is maximal at mid-trimester [46].

## Statistical analysis

All data collected was entered into Microsoft Excel (MS Excel 2016) and cleaned for entry errors. Data was analyzed using IBM-statistical package for social sciences (IBM-SPSS) version 23 (IBM-SPSS, Inc, Chicago, Il, USA). Normally distributed variables were expressed as means and standard deviations (SD) while categorial variables were summarized as numbers and percentages. The wealth index of study respondents was determined using PCA. Before carrying out the analysis, the assumptions were checked, that is; Kaiser-Meyer-Olkin measure of sampling adequacy (KMO > 0.5) and Bartlett's test of sphericity (P < 0.05). Criteria for extraction was eigen value greater than 1 after which varimax (orthogonal) rotation with Kaiser Normalization was used.

Malaria parasite density was log transformed before analysis. Associations between the predictor variables and the primary outcome variables (malaria infection and dietary diversity) was explored using Pearson Chi-square test ($\chi^2$). Differences in group means were compared using Student's t-test, analysis of variance (ANOVA), Mann Whitney U or Kruskall Wahlis test were appropriate. Multivariate logistic regression model was used to determine risk factors associated with malaria parasite infection and dietary diversity. Prior to the analysis, covariates with explanatory plausibility and those with P < 0.2 from the bivariate analysis were subjected to multicollinearity testing using the variance inflation factor (VIF). Collinearity was absent with all covariates having a VIF < 1. Crude odd ratios (COR) were computed using the confidence interval (CI) calculator. Significant levels were measured at 95% CI with significant differences set at P < 0.05.

## Results

### Characteristics of the study participants

A total of 1,014 pregnant women with mean age 26.72 ± 5.48 (range: 15–46) years and gestational age 27.60 ± 7.61 (range: 6–43) weeks were enrolled from Tiko (50.2%, 509/1,014) and Buea (49.8%, 505/1,014) health districts in the Mount Cameroon region. Amongst those enrolled, over 50% were above 25 years, had a low wealth index and were in the third trimester of pregnancy. Moreover, less than a quarter (22.7%) achieved at least four or more clinic visits and only 12.2% took at least three SP doses. Both paucipara and multigravids constituted 43.3% and 40.1% of the study participants respectively. Majority (71.5%) of the women

surveyed owned an insecticide treated net but, only 39.1% made use of it the night prior to enrolment. The overall prevalence of anaemia and undernutrition was; 40.9% and 89.6% respectively as shown in Table 1.

## Malaria prevalence and density

Overall, the prevalence of malaria parasitaemia and malaria anaemia was 17.8% (95% CI: 15.4–20.1; n = 180) and 23.4% (97/415) respectively. Malaria parasite densities ranged from 40–9280 parasites / μL of blood with a geometric mean parasite density (GMPD) of 301.6 ± 1527.5 parasites / μL of blood. A larger proportion of women presented with low parasite density (13.6%, 95% CI: 11.4–15.8; n = 138) while 3.5% (95% CI: 2.4–4.6; n = 35) and 0.7% (95% CI: 0.2–1.3; n = 7) had moderate and high parasitaemia levels respectively. As shown in Table 2, the prevalence of malaria parasitaemia was significantly higher (P = 0.029) among women enrolled during the rainy season (19.1%) when compared with their counterparts enrolled during the dry season (12.7%). Likewise, study respondents who were febrile (25.6%) had a significantly higher (P < 0.001) malaria parasitaemia prevalence when compared with their contemporaries who were afebrile (15.1%). Geometric mean parasite density differed significantly with maternal age (P < 0.001), and gravidity status (P = 0.010) whereby, women aged ≤ 20 years of age and who were primigravids recoded the highest GMPD when compared with their respective counterparts. Also, although not statistically significant GMPD was higher among study participants who did not use their mosquito nets (322 parasites / μL of blood) than in those who made use of it (275 parasites / μL of blood) the night before the survey (Table 2).

## Predictors of infection with malaria parasitaemia

To investigate the risk factors of malaria parasite infection among study participants, the multivariate logistic regression modelling was performed. After adjusting for possible confounders, the model revealed season (P = 0.043), being a farmer (P = 0.030), a family size less than 4 persons (P = 0.026) and fever (P < 0.001) as significant predictors of malaria parasite infection. Considering the odds ratios, respondents who were enrolled in the rainy season (OR = 1.58, 95% CI: 1.01–2.48), engaged in farming (OR = 2.30, 95% CI: 1.08–4.88) and had a household size of < 4 individuals (OR = 1.48, 95% CI: 1.04–2.11) were at increased odds of being infected with malaria parasites when compared with their respective contemporaries. Also, women who had fever were 1.87-folds more likely to be infected with malaria parasites when compared with their counterparts who did not present with fever during enrollment into the study. Although not statistically significant, women who were undernourished were at increased risk (OR = 1.56, 95% CI: 0.82–2.96) of being infected with malaria parasites (Table 3).

## Factors associated with dietary diversity

In the unadjusted analysis as shown in Table 4, health facility type, maternal age and number of clinic visits attended were significantly (P < 0.05) associated with undernutrition. Variables with explanatory plausibility and P < 0.2 from the bivariate analysis were run in the multivariate logistic regression analysis in order to ascertain predictors of poor dietary diversity. The analysis showed that, women attending antenatal care at THHC (OR = 2.18, 95% CI: 0.91–5.21), MMC (OR = 1.94, 95% CI: 1.12–3.35) and BIHC (OR = 2.93, 95% CI: 1.66–5.20) were 2.2, 1.9 and 2.9 times more likely to have an inappropriate diet than those enrolled from MMH. Furthermore, compared with older women (>25 years), those ≤ 20 years (OR = 2.12, 95% CI: 0.94–0.47) and within the age group 21–25 years (OR = 2.43, 95% CI: 1.38–4.30) were 2.1 and 2.4 times at increased odds of having a less diverse diet. The likelihood of a poor

**Table 1. Characteristics of the study participants.**

| Parameter | Category | Value % (n) |
|---|---|---|
| Season at enrollment | Dry | 21.0 (213) |
| | Rainy | 79.0 (801) |
| Health facility type | THHC | 7.8 (79) |
| | MMC | 42.4 (430) |
| | BIHC | 33.7 (342) |
| | MMH | 16.1 (163) |
| Age (years) | $\leq 20$ | 14.3 (145) |
| | 21–25 | 29.1 (295) |
| | >25 | 56.6 (574) |
| Marital status | Single | 37.8 (383) |
| | Married | 62.2 (631) |
| Educational level | Primary | 20.1 (204) |
| | Secondary | 53.1 (538) |
| | Tertiary | 26.8 (272) |
| Occupation | Housewife | 16.5 (167) |
| | Farmer | 5.8 (59) |
| | Business | 43.6 (442) |
| | Student | 19.5 (198) |
| | Civil servant | 14.6 (148) |
| ANC visits attended | 1 | 30.8 (312) |
| | 2 | 27.6 (280) |
| | 3 | 18.9 (192) |
| | $\geq 4$ | 22.7 (230) |
| Parity | Nullipara (0) | 38.5 (390) |
| | Paucipara (1–2) | 43.3 (439) |
| | Multipara (3–4) | 16.5 (167) |
| | Grand multipara ($\geq 5$) | 1.8 (18) |
| Gravidity | Primigravid | 34.6 (351) |
| | Secundigravid | 25.2 (256) |
| | Multigravid | 40.1 (407) |
| Trimester of pregnancy | First | 3.5 (35) |
| | Second | 41.2 (418) |
| | Third | 55.3 (561) |
| IPTp-SP dosage frequency | 0 | 41.9 (425) |
| | 1 | 25.7 (261) |
| | 2 | 20.1 (204) |
| | $\geq 3$ | 12.2 (124) |
| IFA uptake | Yes | 71.5 (725) |
| | No | 28.5 (289) |
| ITN ownership | Yes | 71.2 (722) |
| | No | 28.8 (292) |
| ITN usage | Yes | 39.1 (396) |
| | No | 60.9 (618) |
| Fever status | Febrile | 25.0 (254) |
| | Afebrile | 75.0 (760) |
| Family size | < 4 | 38.2 (387) |
| | $\geq 4$ | 61.8 (627) |

(*Continued*)

**Table 1.** (Continued)

| Parameter | Category | Value % (n) |
|---|---|---|
| Wealth index | Low | 56.8 (576) |
| | High | 43.2 (438) |
| Anaemia status | Anaemic | 40.9 (415) |
| | Non anaemic | 59.1 (599) |
| MDD-W status | Inadequate dietary diversity | 89.6 (909) |
| | Adequate dietary diversity | 10.4 (105) |
| **Means** | **Mean (± SD)** | **Range** |
| Mean age in years | 26.72 ± 5.48 | 15–46 |
| Mean ANC visits | 2.54 ± 1.58 | 1–12 |
| Mean GA in weeks | 27.60 ± 7.61 | 6–43 |
| Mean temperature in ˚C | 36.52 ± 0.64 | 35.0–39.0 |
| Mean Hb levels in g/dL | 11.00 ± 1.34 | 7.7–14.3 |
| Mean MDD-W score | 3.57 ± 0.82 | 1–7 |

THHC, Tiko Holforth Health Centre; MMC, Mutengene Medical Centre; BIHC, Buea Integrated Health Centre; MMH, Mount Mary Hospital; ANC, antenatal clinic; intermittent preventive treatment in pregnancy with sulphadoxin-pyrimethamine, IPTp-SP; IFA, iron folic acid; ITN, insecticide treated net; MDD-W, minimum dietary diversity for women; SD, standard deviation.

dietary intake was 8.8 and 1.6 times higher among women engaged in farming (OR = 8.86, 95% CI: 0.41–1.81) and in those who attended < 4 clinic visits (OR = 1.60, 95% CI: 1.01–2.55) when compared with their respective counterparts (Table 4).

## Mean dietary diversity scores with respect to maternal socio-demographic and antenatal characteristics

As shown in Table 5, mean dd scores were significantly (P < 0.05) higher among pregnant women above 25 years of age (3.64 ± 0.841), who were enrolled from the Mount Mary Hospital (3.79 ± 0.864) and who attended at least four or more clinic visits (3.78 ± 0.818) when compared with their contemporaries respectively. Also, mean dd scores were significantly (P = 0.047) higher among those without fever (3.60 ± 0.807) when compared with their counterparts who presented with fever (3.48 ± 0.865).

## Influence of independent variables on dietary diversity scores

In Table 6 below, multilinear regression (enter) modelling was run to assess the influence of the various independent variables on dd (dietary diversity) scores. The analysis showed that maternal age (P = 0.011) and number of clinic visits attended (P < 0.001) significantly influenced dd scores (Table 6).

## Association between malaria infection and dietary diversity

The mean MDD-W score was lower in women positive for infection with malaria parasites (3.43 ± 0.799; score range: 1–6) when compared with their counterparts who were negative (3.60 ± 0.826; score range: 1–7). This association was statistically significant (t = -2.444; P = 0.015). As shown in Fig 1, the MDD-W score was significantly (F = 3.823; P = 0.024) lower among pregnant women who harboured high malaria parasitaemia levels (2.71 ± 1.113; score range 1–4) when compared with their equivalents who had moderate (3.31 ± 0.796; score range: 1–5) and low malaria parasite levels (3.50 ± 0.767; score range: 1–6). The overall

**Table 2. Malaria prevalence and geometric mean parasite density by season, socio-demographic and antenatal characteristics of the study participants.**

| Parameter | Category | Number examined | Prevalence % (n) | P value | GMPD (range)/ μL of blood | P value |
|---|---|---|---|---|---|---|
| Season at enrolment | Dry | 213 | 12.7 (27) | **0.029**[*] | 197 (40–5080) | **0.022**[a*] |
| | Rainy | 801 | 19.1 (153) | | 325 (80–9280) | |
| Setting | THD | 509 | 18.1 (92) | 0.787 | 324 (40–8400) | 0.406[a] |
| | BHD | 505 | 17.4 (88) | | 279 (40–9280) | |
| Age (years) | ≤ 20 | 145 | 24.1 (35) | 0.092 | 630 (80–8400) | **< 0.001**[b*] |
| | 21–25 | 295 | 16.3 (48) | | 284 (40–9280) | |
| | >25 | 574 | 16.9 (97) | | 238 (40–8400) | |
| Education level | Primary | 204 | 20.6 (42) | 0.478 | 220 (40–2600) | **0.013**[b*] |
| | Secondary | 538 | 17.3 (93) | | 361 (40–8400) | |
| | Tertiary | 272 | 16.5 (45) | | 279 (80–9280) | |
| Marital status | Single | 383 | 18.0 (69) | 0.864 | 323 (40–8400) | 0.524[a] |
| | Married | 631 | 17.6 (111) | | 289 (40–9280) | |
| Gravidity | Primigravid | 351 | 18.5 (65) | 0.775 | 423 (80–9280) | **0.010**[b*] |
| | Secundigravid | 256 | 18.4 (47) | | 277 (80–6160) | |
| | Multigravid | 407 | 16.7 (68) | | 231 (40–8400) | |
| No of clinic visits | 1 | 312 | 17.6 (55) | 0.997 | 334 (80–8400) | 0.229[b] |
| | 2 | 280 | 17.5 (49) | | 281 (40–5040) | |
| | 3 | 192 | 18.2 (35) | | 325 (80–6000) | |
| | ≥ 4 | 230 | 17.8 (41) | | 269 (40–9280) | |
| IPTp-SP dosage frequency | 0 | 425 | 18.4 (78) | 0.490 | 334 (40–8400) | 0.102[b] |
| | 1 | 261 | 18.4 (48) | | 293 (40–8400) | |
| | 2 | 204 | 14.2 (29) | | 348 (80–9280) | |
| | ≥ 3 | 124 | 20.2 (25) | | 196 (80–1600) | |
| ITN usage | Yes | 396 | 18.9 (75) | 0.428 | 275 (40–6160) | 0.277[a] |
| | No | 618 | 17.0 (105) | | 322 (80–9280) | |
| Fever status | Febrile | 254 | 25.6 (65) | **< 0.001**[*] | 515 (80–9280) | **< 0.001**[a] |
| | Afebrile | 760 | 15.1 (115) | | 223 (40–2600) | |

THD, Tiko Health District; BHD, Buea Health District; intermittent preventive treatment in pregnancy with sulphadoxin-pyrimethamine, IPTp-SP; ITN, insecticide treated net.

[*]Statistically significant at P <0.05.

[a] Difference in GMPD determined by Mann Whitney *U* test.

[b] Difference in GMPD determined by Kruskal-Wallis test.

prevalence of fever was 25% (95% CI: 22.4–27.9; n = 254). Among those with fever, 6.4% (95% CI: 4.8–8.0; n = 65) and 18.6% (95% CI: 16.3–21.2; n = 189) presented with current fever and a history of fever respectively. Also, of the total women enrolled, 6.4% (95% CI: 4.9–8.0; n = 65) presented with malaria and fever whereas 11.3% (95% CI: 9.5–13.3; n = 115) had malaria without fever. The MDD-W score was significantly lower (F = 4.198, P = 0.006) among study respondents presenting with symptomatic malaria infection (3.23 ± 0.786; score range: 1–5), asymptomatic malaria infection (3.55 ± 0.786; score range: 1–6), and fever alone (3.57 ± 0.876; score range: 1–6) when compared with their counterparts who presented with none of the above morbidities (3.61 ± 0.811; score range: 1–7) (Fig 2).

## Discussion

In low-income settings, malaria and undernutrition are major health challenges in pregnancy contributing significantly to unfavorable outcomes [3,4,11]. Therefore, this study aimed to

**Table 3. Logistic regression analysis showing predictors of malaria parasite infection status.**

| Variable | N | Malaria positive % (n) | Bivariate logistic regression | Multivariate logistic regression | |
|---|---|---|---|---|---|
| | | | COR (95% CI) | AOR (95% CI) | P value |
| **Season** | | | | | |
| Rainy | 801 | 19.1 (153) | 1.62 (1.04–2.52) | 1.58 (1.01–2.48) | **0.043**\* |
| Dry | 213 | 12.7 (27) | **REFERENCE** | **REFERENCE** | |
| **Health facility type** | | | | | |
| THHC | 79 | 13.9 (11) | 0.74 (0.35–1.58) | - | - |
| MMC | 430 | 18.8 (81) | 1.07 (0.67–1.71) | - | - |
| BIHC | 342 | 17.3 (59) | 0.96 (0.59–1.57) | - | - |
| MMH | 163 | 17.8 (29) | **REFERENCE** | - | - |
| **Age (years)** | | | | | |
| ≤ 20 | 145 | 24.1 (35) | 1.56 (1.00–2.42) | 1.50 (0.84–2.67) | 0.163 |
| 21–25 | 295 | 16.3 (48) | 0.95 (0.65–1.39) | 0.92 (0.59–1.44) | 0.741 |
| >25 | 574 | 16.9 (97) | **REFERENCE** | **REFERENCE** | |
| **Education level** | | | | | |
| Primary | 204 | 20.6 (42) | 1.30 (0.82–2.08) | - | - |
| Secondary | 538 | 17.3 (93) | 1.05 (0.71–1.55) | - | - |
| Tertiary | 272 | 16.5 (45) | **REFERENCE** | - | - |
| **Marital status** | | | | | |
| Single | 383 | 18.0 (69) | 1.03 (0.73–1.43) | - | - |
| Married | 631 | 17.6 (111) | **REFERENCE** | - | - |
| **Occupation** | | | | | |
| Housewife | 167 | 15.0 (25) | 1.00 (0.54–1.87) | 0.83 (0.43–1.61) | 0.600 |
| Farmer | 59 | 28.8 (17) | 2.31 (1.12–4.77) | 2.30 (1.08–4.88) | **0.030**\* |
| Business | 442 | 17.4 (77) | 1.20 (0.72–2.02) | 1.08 (0.63–1.84) | 0.768 |
| Student | 198 | 19.7 (39) | 1.40 (0.79–2.49) | 1.11 (0.58–2.13) | 0.746 |
| Civil servant | 148 | 14.9 (22) | **REFERENCE** | **REFERENCE** | |
| **Clinic visits attended** | | | | | |
| < 4 visits | 784 | 17.7 (139) | 0.99 (0.67–1.45) | - | - |
| ≥ 4 visits | 230 | 17.8 (41) | **REFERENCE** | - | - |
| **Gravidity status** | | | | | |
| Primigravid | 351 | 18.5 (65) | 1.13 (0.77–1.64) | 1.01 (0.55–1.57) | 0.806 |
| Secundigravid | 256 | 18.4 (47) | 1.12 (0.74–1.68) | 1.00 (0.63–1.59) | 0.975 |
| Multigravid | 407 | 16.7 (68) | **REFERENCE** | **REFERENCE** | |
| **IPTp-SP uptake** | | | | | |
| No | 426 | 18.3 (78) | 0.93 (0.67–1.29) | 1.00 (0.71–1.39) | 0.996 |
| Yes | 588 | 17.3 (102) | **REFERENCE** | **REFERENCE** | |
| **ITN usage** | | | | | |
| No | 618 | 17.0 (105) | **REFERENCE** | - | - |
| Yes | 396 | 18.9 (75) | 1.06 (0.77–1.47) | - | - |
| **Family size** | | | | | |
| < 4 | 387 | 20.9 (81) | 1.41 (1.01–1.95) | 1.48 (1.04–2.11) | **0.026**\* |
| ≥ 4 | 627 | 15.8 (99) | **REFERENCE** | **REFERENCE** | |
| **Wealth index** | | | | | |
| Low | 576 | 18.2 (105) | 1.07 (0.77–1.49) | - | - |
| High | 438 | 17.1 (75) | **REFERENCE** | - | - |
| **Fever status** | | | | | |

*(Continued)*

**Table 3.** (Continued)

| Variable | N | Malaria positive % (n) | Bivariate logistic regression | Multivariate logistic regression | |
|---|---|---|---|---|---|
| | | | COR (95% CI) | AOR (95% CI) | P value |
| Yes | 254 | 25.6 (65) | 1.92 (1.36–2.72) | 1.87 (1.31–2.67) | <**0.001**$^*$ |
| No | 760 | 15.1 (115) | **REFERENCE** | **REFERENCE** | |
| Dietary diversity status | | | | | |
| Inadequate dd | 909 | 18.5 (168) | 1.75 (0.94–3.27) | 1.56 (0.82–2.96) | 0.168 |
| Adequate dd | 105 | 11.4 (12) | **REFERENCE** | **REFERENCE** | |

THHC, Tiko Holforth Health Centre; MMC, Mutengene Medical Centre; BIHC, Buea Integrated Health Centre; MMH, Mount Mary Hospital; intermittent preventive treatment in pregnancy with sulphadoxin-pyrimethamine, IPTp-SP; ITN, insecticide treated net; dd, dietary diversity; COR, crude odds ratio; AOR, adjusted odds ratio
$^*$Statistically significant at P <0.05.

assess the association between malaria infection and poor nutrition among pregnant women in the study area.

In the present study, the overall prevalence of malaria was 17.8%. This prevalence rate is comparable with 16.0% obtained in the study area [3] and 16.1% in Burkina Faso [47] but lower than that documented in Foumban (53.4%) [48]. Difference in study design might explain this variation. For instance, the study in Foumban [48] was population based whereas our study was a hospital based cross sectional survey. Also, poor environmental conditions such as; presence of bushes and standing water might as well explain these disparities. On the other hand, the present finding is higher than 13.4% recorded in the same study setting [38]. Earlier studies of IPTp in Cameroon demonstrated a clear reduction in parasitaemia following receipt of SP [49,50]. Thus, the relatively high prevalence of infection with malaria parasitaemia may be due to suboptimal ITN use and IPTp-SP uptake. At the time of the study, IPTp-SP coverage of at least one dose was 67.6% (686/1,014) which however, is lower compared with 90.5% reported in a previous study in the Mount Cameroon area [51]. In addition, only 39.1% of the women reported usage of ITN. More so, inadequate uptake of malaria preventive measures may be attributed to inadequate ANC clinic attendance or late ANC initiation [51].

In agreement with reports of several studies, malaria parasite levels decreased with increasing maternal age and gravidity status [1,3,49,52]. This might be linked to pregnancy-associated anti-parasite immunity. The variant surface antigen (VAR2CSA), a specific variant of *P. falciparum* erythrocyte membrane protein 1 (PfEMP1) is expressed on the surface of infected erythrocytes (IEs) [53–55]. This expression mediates adhesion to chondroitin sulfate A (CSA) [56] thus, enabling sequestration of IEs in the placenta [57]. Primigravidae and secundigravidae are more prone to malaria as they do not possess significant levels of VAR2CSA-specific IgG which is acquired only after successive pregnancy and with increasing maternal age [58]. Furthermore, this study equally revealed that malaria parasitaemia varied significantly with maternal education level. Similar observations have been reported elsewhere [59,60]. Low maternal education levels may contribute to inadequate understanding of malaria preventive measures which in turn increases the risk of adverse pregnancy outcomes due to malaria parasite infection. Thus, government policies and programme initiatives should target improving maternal education statuses as this would be vital in reducing the burden of malaria among this vulnerable group.

Analysis of the factors associated with malaria infection demonstrated that climatic season at enrolment, being a farmer, having a household number of less than four individuals and the presence of fever were significantly associated with malaria. In line with the current study,

**Table 4. Logistic regression analysis showing predictors of dietary diversity.**

| Factor | Category | N | PDD % (n) | Bivariate logistic regression | | Multivariate logistic regression | |
|---|---|---|---|---|---|---|---|
| | | | | COR (95% CI) | P value | AOR (95% CI) | P value |
| Season | Dry | 213 | 91.5 (195) | 1.32 (0.77–2.24) | 0.376 | - | - |
| | Rainy | 801 | 89.1 (714) | **REFERENCE** | | - | - |
| Health facility type | THHC | 79 | 89.9 (71) | 2.16 (0.94–4.95) | **< 0.001*** | 2.18 (0.91–5.21) | 0.078 |
| | MMC | 430 | 91.2 (392) | 2.51 (1.51–4.19) | | 1.94 (1.12–3.35) | **0.017*** |
| | BIHC | 342 | 92.1 (315) | 2.84 (1.64–4.94) | | 2.93 (1.66–5.20) | **< 0.001*** |
| | MMH | 163 | 80.4 (131) | **REFERENCE** | | **REFERENCE** | |
| Age (years) | ≤ 20 | 145 | 93.1 (135) | 2.06 (1.03–4.09) | **0.003*** | 2.12 (0.94–0.47) | 0.070 |
| | 21–25 | 295 | 93.6 (276) | 2.21 (1.31–3.74) | | 2.43 (1.38–4.30) | **0.002*** |
| | >25 | 574 | 86.8 (498) | **REFERNCE** | | **REFERENCE** | |
| Marital status | Single | 383 | 90.3 (346) | 1.12 (0.74–1.72) | 0.572 | - | - |
| | Married | 631 | 89.2 (563) | **REFERENCE** | | - | - |
| Education level | Primary | 204 | 91.2 (186) | 1.52 (0.83–2.78) | 0.267 | - | - |
| | Secondary | 538 | 90.3 (486) | 1.38 (0.87–2.17) | | - | - |
| | Tertiary | 272 | 87.1 (237) | **REFERENCE** | | - | - |
| Occupation | Housewife | 167 | 88.6 (148) | 1.28 (0.66–2.50) | 0.106 | 0.86 (0.41–1.81) | 0.710 |
| | Farmer | 59 | 98.3 (58) | 9.59 (1.25–73.02) | | 8.86 (1.07–72.83) | **0.042*** |
| | Business | 442 | 86.6 (396) | 1.42 (0.81–2.47) | | 1.06 (0.55–2.06) | 0.845 |
| | Student | 198 | 90.9 (180) | 1.65 (0.84–3.22) | | 0.99 (0.45–2.14) | 0.983 |
| | Civil servant | 148 | 85.8 (127) | **REFERENCE** | | **REFERENCE** | |
| Clinic visits attended | < 4 | 784 | 91.1 (714) | 1.83 (1.18–2.83) | **0.006*** | 1.60 (1.01–2.55) | **0.044*** |
| | ≥ 4 | 230 | 84.8 (195) | **REFERENCE** | | **REFERENCE** | |
| Family size | < 4 | 387 | 91.7 (355) | 1.46 (0.94–2.26) | 0.087 | 1.39 (0.88–2.18) | 0.150 |
| | ≥ 4 | 627 | 88.4 (554) | **REFRENCE** | | **REFERENCE** | |
| Wealth index | Low | 576 | 90.1 (519) | 1.12 (0.74–1.68) | 0.582 | - | - |
| | High | 438 | 89.0 (390) | **REFERENCE** | | - | - |
| Fever status | Febrile | 254 | 91.7 (233) | 1.37 (0.83–2.27) | 0.207 | 1.38 (0.82–2.31 | 0.217 |
| | Afebrile | 760 | 88.9 (676) | **REFERENCE** | | **REFERENCE** | |

PDD, poor dietary diversity; THHC, Tiko Holforth Health Centre; MMC, Mutengene Medical Centre; BIHC, Buea Integrated Health Centre; MMH, Mount Mary Hospital; COR, crude odds ratio; AOR, adjusted odds ratio
*Statistically significant at P <0.05.

previous studies have reported an association between climatic season particularly the rainy season with increased malaria infection transmission [61,62]. It is well known that rainfall plays an important role in malaria epidemiology because water not only provides breeding sites for mosquitoes but also increases the longevity of the adult mosquitoes [61,63]. Likewise, the risk of infection with malaria has been documented to be linked with the occupation of an individual [64]. Results from this study showed that farmers were 2.3 times at increased odds of becoming infected with malaria parasites. Similar observations were found in studies carried out in Ethiopia [65], Ghana [66], Kenya [67] and Congo [68]. This could be because these women practiced agricultural activities (a form of outdoor activity) either at dawn or dusk thus increasing the risk of them receiving infective bites from the mosquito vector [69]. Moreover, reports from several studies have persistently shown an increased risk of malaria infection among individuals living in houses with many family members [68,70,71]. Study findings however demonstrated otherwise that pregnant women in households of one to three family

**Table 5. Mean DD scores with respect to maternal socio-demographic and antenatal characteristics.**

| Factor | Categories | Mean ±SD | P value |
|---|---|---|---|
| Season | Dry | 3.52 ± 0.804 | 0.294[a] |
| | Rainy | 3.58 ± 0.828 | |
| Health facility type | THHC | 3.71 ± 0.754 | < **0.001**[b]* |
| | MMC | 3.55 ± 0.826 | |
| | BIHC | 3.46 ± 0.794 | |
| | MMH | 3.79 ± 0.864 | |
| Age (years) | < 20 | 3.46 ± 0.866 | **0.004**[b]* |
| | 21–25 | 3.47 ± 0.750 | |
| | >25 | 3.64 ± 0.841 | |
| Education level | Primary | 3.60 ± 0.753 | **0.020**[b]* |
| | Secondary | 3.51 ± 0.831 | |
| | Tertiary | 3.67 ± 0.850 | |
| Occupation | Housewife | 3.61 ± 0.849 | 0.377[b] |
| | Farmer | 3.47 ± 0.653 | |
| | Business | 3.55 ± 0.824 | |
| | Student | 3.52 ± 0.811 | |
| | Civil servant | 3.67 ± 0.868 | |
| Clinic visits attended | < 4 | 3.51 ± 0.815 | **0.001**[a]* |
| | ≥ 4 | 3.78 ± 0.818 | |
| Family size | < 4 | 3.51 ± 0.816 | 0.068[a] |
| | ≥ 4 | 3.61 ± 0.827 | |
| Fever status | Febrile | 3.48 ± 0.865 | **0.047**[a]* |
| | Afebrile | 3.60 ± 0.807 | |

THHC, Tiko Holforth Health Centre; MMC, Mutengene Medical Centre; BIHC, Buea Integrated Health Centre; MMH, Mount Mary Hospital; ANC, antenatal clinic; SD, standard deviation

*Statistically significant at P < 0.05.

[a] means compared using student T-test.

[b] means compared using one-way ANOVA.

**Table 6. MLR analysis examining the influence of independent variables on dietary diversity scores.**

| Factor | B | 95% CI | P value |
|---|---|---|---|
| Season at enrolment | 0.047 | -0.077–0.172 | 0.457 |
| Health facility type | 0.013 | -0.049–0.079 | 0.670 |
| Age | 0.091 | 0.021–0.162 | **0.011*** |
| Education | 0.045 | -0.036–0.127 | 0.276 |
| Occupation | -0.009 | -0.053–0.035 | 0.676 |
| Clinic visits attended | 0.257 | 0.137–0.378 | < **0.001*** |
| Family size | 0.085 | -0.019–0.190 | 0.109 |
| Fever status | 0.105 | -0.011–0.222 | 0.076 |

MLR, multiple linear regression; B, unstandardized beta

*Statistically significant at P < 0.05.

MLR Model summary: R = 0.187, $R^2$ = 0.035, Adjusted $R^2$ = 0.027, F = 4.574, P < 0.001.

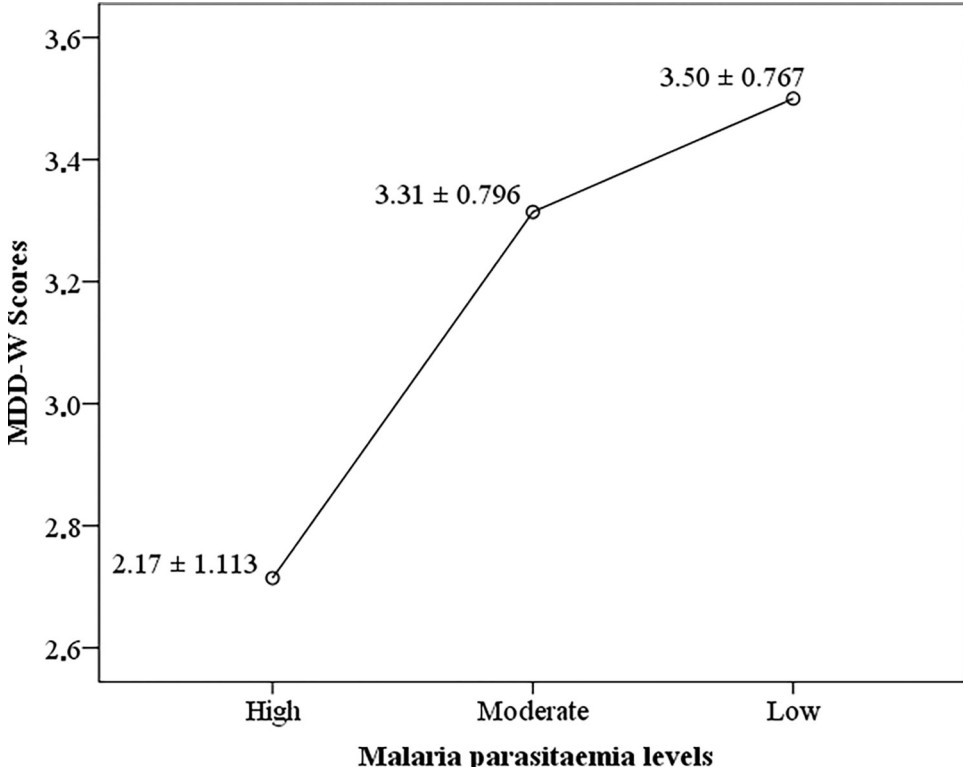

**Fig 1. Dietary diversity scores Vs malaria parasitaemia levels.**

members were 1.4 times at increased odds of being infected with malaria. The increased risk of infection among members in households with low occupancy might be due to the fact that they were significantly younger ($\leq$ 25 years) and the majority were primigravidae. Infection with malaria parasites often presents with a vast array of symptoms one of which is fever. Although fever is a fairly sensitive indicator of malaria especially in settings where infection with other pathogens are possible [72] in this study, it was significantly associated with an increased risk of infection among pregnant women. Similar significant association between malaria and fever has been documented in the study area [3,49,73].

With regards to poor dietary diversity, the overall prevalence was 89.6%. Besides, we observed health facility type, inadequate antenatal visit attendance, younger maternal age and farming as an occupation as prominent factors associated with poor nutrition among pregnant women. Nutrition interventions during pregnancy can be delivered through various platforms; among them, healthcare systems are considered the most effective platform to reach and counsel expectant mothers on adequate dietary practices as well as to prepare them for breastfeeding [9,74]. According to the present study, the likelihood of having a diverse diet was higher among women attending clinic visits in THHC and MMH than those from BIHC and MMC. This finding suggests gaps in the quality of nutrition counselling offered by the different health facilities in the Mount Cameroon region. An effective ANC package depends on competent health care providers in a functioning health system with referral services and adequate supplies and laboratory support [75]. Sub-optimal ANC (< 4 visits) was positively (OR = 1.6) associated with uptake of inadequate nutrient diet and corroborates reports by other authors [76,77]. Pregnant women may have missed scheduled ANC opportunities to be educated on healthy dietary lifestyle during pregnancy. The anglophone crisis in the English-speaking

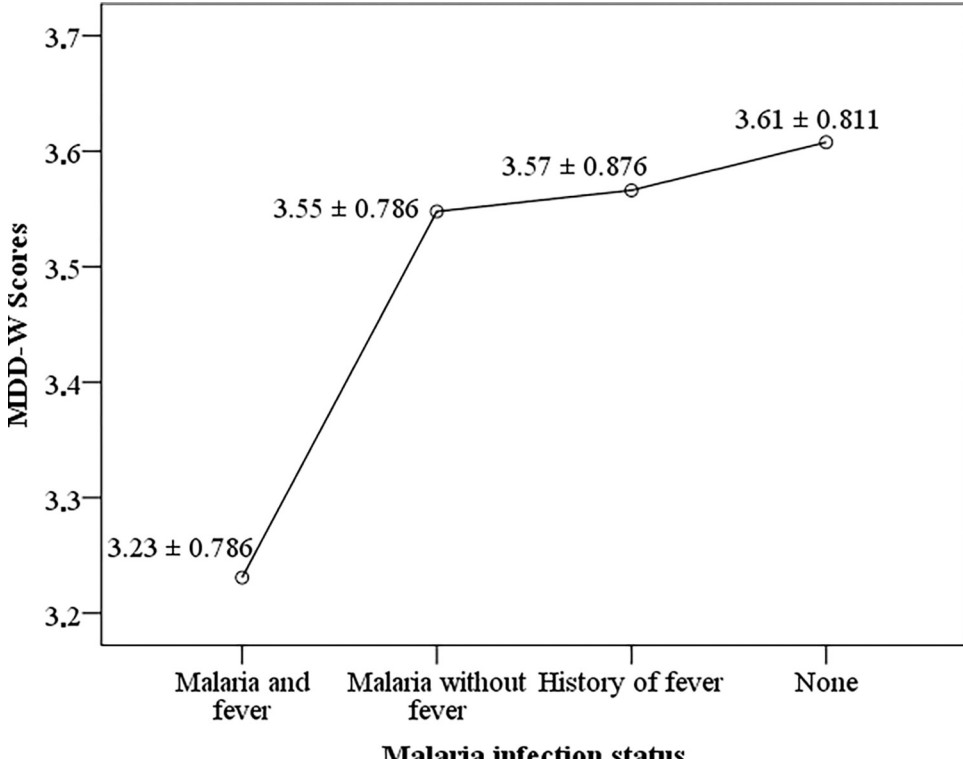

**Fig 2. Dietary diversity scores Vs malaria infection status.**

regions of Cameroon since 2017 [26] and/or late ANC initiation [51] may have hampered ANC clinic utilization. Exposure to high prenatal visits and nutrition information from qualified health professionals are more likely to enhance good dietary practice. Nutrition information in the form of dietary counselling during clinic visit plays a major role in improving nutrient intake [78,79] and nutritional status as well [80,81].

Congruent to findings in Ethiopia [82] women engaged in agricultural activities in the study area were 8.8 times more likely to have a less diverse diet. The increased odds of dietary inadequacy among these women could be that they do not consume what they produce as they rather sell farm products to generate income to support livelihood of the family [83,84]. Poor dietary intake among farmers may also be related to the fact that most of them had a low level of education (46 out of 59). Higher levels of education are associated with healthier dietary pattern [85]. Study findings showed that the odds of poor DD increased with younger age; women ≤ 25 years old had more than two-fold risks of having a less diverse diet when compared with those over 25 years. This finding is supported by similar studies [86–88]. Older women are more aware of the importance of a diverse diet in achieving optimum health during pregnancy from previous experience [89]. Moreover, younger women may have limited knowledge, socioeconomic and household support [90].

Consistent with the finding of previous studies [27,91–93], we observed a significantly low DD score among women infected with malaria parasitaemia whereas others found no association between these two co-morbidities [94–96]. It is well established that undernutrition supresses the body's immune response [97,98] thus, exposing the individual to infections like malaria [93,99,100]. Likewise, the nutrition of women infected with malaria parasitaemia may have been worsened by loss of appetite as well as the diversion of dietary nutrients for immune response.

This study has several limitations. Firstly, the cross-sectional nature cannot establish the cause-and-effect relationship between the predictors and outcome variables. Secondly, there might be recall bias as study participants had to remember the food which they consumed a day before the survey. Also, the high confidence interval level observed in maternal occupation (farmer) as a predictor of dietary diversity is likely due to the small number of women recorded to be engaged in farming. Even though this study employed the use of microscopy which is the gold standard for malaria diagnosis, the prevalence of malaria may have been under estimated considering that sub-microscopic infections are common in the study area. More so, this study did not evaluate the influence of sub-microscopic malaria infection and soil-transmitted helminths on maternal nutrition. Nevertheless, the findings of the study demonstrated the main determinants of two major health problems in our setting in the context of attaining the SDGs. Also, this study provides the basis for further studies on the association between maternal nutrition and the risk of malaria infection.

## Conclusion

In summary, 17.8% of the respondents were infected with malaria parasites. Among those infected, GMPD was 301 parasites/μL of blood (range: 40–9280). Infection with malaria parasites was significantly associated with climatic season at enrollment, occupation, family size and fever status whereas health facility type, younger maternal age, farming as a form of occupation and inadequate clinic visits were factors that significantly increased the prevalence of poor dietary nutrient intake. Besides, the association between malaria and dietary diversity was statistically significant. Since nutrition is tied to immune response, there is need for improving awareness through multi-sectoral collaboration on the benefits of consuming at least five or more food groups among pregnant women. Likewise, there should be integration of nutrition activities with malaria control programs in order to efficiently manage malaria among pregnant Cameroonian women.

## Supporting information

**S1 Dataset.**
(XLSX)

## Acknowledgments

The authors would like to thank the administrative staff, nurses and laboratory technicians of the various health facilities where this study was carried out for their assistance and collaboration.

## Author Contributions

**Conceptualization:** Vanessa Tita Jugha, Judith Kuoh Anchang-Kimbi.

**Data curation:** Vanessa Tita Jugha.

**Formal analysis:** Vanessa Tita Jugha, Juliana Adjem Anchang, Germain Sotoing Taiwe, Judith Kuoh Anchang-Kimbi.

**Supervision:** Helen Kuokuo Kimbi, Judith Kuoh Anchang-Kimbi.

**Validation:** Helen Kuokuo Kimbi, Judith Kuoh Anchang-Kimbi.

**Writing – original draft:** Vanessa Tita Jugha.

**Writing – review & editing:** Juliana Adjem Anchang, Germain Sotoing Taiwe, Helen Kuokuo Kimbi, Judith Kuoh Anchang-Kimbi.

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
