## [Decision Letter · Decision Letter 0]

14 Mar 2023

PONE-D-23-01027Association between malaria and inadequate dietary diversity score among pregnant women at presentation for antenatal care in health facilities in the Mount Cameroon regionPLOS ONE

Dear Dr. Jugha,

Thank you for submitting your manuscript to PLOS ONE. After careful consideration, we feel that it has merit but does not fully meet PLOS ONE’s publication criteria as it currently stands. Therefore, we invite you to submit a revised version of the manuscript that addresses the points raised during the review process.

ACADEMIC EDITOR:Kindly revise you manuscript taking into consideration the comments of both reviewers. Particular attention should please be placed on the statistical analysis conducted and the results presented.

We look forward to receiving your revised manuscript.

Kind regards,

Gifty Dufie Ampofo, M.D., Ph.D

Academic Editor

PLOS ONE

Reviewers' comments:

Reviewer's Responses to Questions

**Comments to the Author**

1. Is the manuscript technically sound, and do the data support the conclusions?

Reviewer #1: Yes

Reviewer #2: Partly

2. Has the statistical analysis been performed appropriately and rigorously? 

Reviewer #1: Yes

Reviewer #2: No

3. Have the authors made all data underlying the findings in their manuscript fully available?

Reviewer #1: Yes

Reviewer #2: Yes

4. Is the manuscript presented in an intelligible fashion and written in standard English?

Reviewer #1: Yes

Reviewer #2: Yes

5. Review Comments to the Author

Reviewer #1: REVIEWER’S COMMENT

Abstract

Point 1: Result for Maternal Dietary Diversity Score needs to be reflected in the abstract.

Point 2: Line 39: the word “conversely” (used for reverse relationship between two statements) is a misfit. “Also” can be used instead.

Point 3: Line 42: … were identified as risk factors of DD… I suggest you omit “risk factors” and use “predictors” instead.

Point 4: There was no conclusion for the abstract.

Point 5: keywords were not listed.

Introduction

Point 6: Line 76-77: … furthermore, the consequences of inadequate maternal nutrient intake for the newborns are equally serious … Please rephrase as “… furthermore, the consequences of inadequate maternal nutrient intake are severe…”.

Point 7: Line 80 -81: Please explain further why anthropometry is often used in low income settings to measure maternal undernutrition.

Methods

Point 8: Line 106-108 and 114-116 (the longitudes and latitudes) should be omitted.

Point 9: Line 123-126 can be can be added to the introduction.

Point 10: Line 135-137 should form part of the ethics statement.

Point 11: Data sampling technique was not stated.

Point 12: Some methodological queries including validation of data collection tools like the questionnaire, the constitution of the team that performed the data collection, training provided to the team were not addressed.

Point 13: Why was marital status not included as it would have peradventure been one of the predictors for inadequate dietary diversity score.

Point 14: Since 24-hour dietary recall was used to assess the maternal dietary diversity, elaborate more on how it was obtained (two weekdays and one weekend).

Point 15: Please state the food groups for readers to have a fair idea about it.

Point 16: Line 148-158: Socio-demographic data should be separated from the clinical assessment. Line 170-176 can form part of the socio-demographic data.

Result

Point17: For table 1, the percentages for both adequate and inadequate dietary diversity for the MMD-W status must be stated. Same applies to the anaemia status, IFA intake, ITN ownership and ITN usage.

Discussion

Point 18: It was stated that irrespective of the level of education, pregnant women in the Mount Cameroon are all exposed to almost the same level of malaria transmission but also stated (line 369-370) that malaria parasitaemia varied significantly with maternal education level. Please clarify the statement.

Conclusion

Point19: State whether there was an association between the predictors of malaria infection and inadequate dietary diversity since study was establish a relation between the prevalence and predictors of malaria infection and inadequate

Reviewer #2: 1) The aim of the study should be synchronized in the abstract, introduction, discussion, as well as with the title of the study. In addition, the undernutrition should be consistently referred to. In the title, it is referred to inadequate dietary diversity score, in the abstract it is referred to as undernutrition, and in the introduction it is inadequate dietary intake. However, reference 24 in this study shows that findings on dietary diversity score were already published. So it is better that as the aim is synchronized, this part is omitted to avoid dual publication

2) The results on prevalence of undernutrition are missing from the abstract. If this is still part of the aims, include them

3) The sample size estimation presented has been shown to be based on prevalence of anemia. However, it should be shown whether the sample size is adequate to answer the objectives of the current study. In addition, the estimation is for descriptive study yet the study has analytical component (the associations). Should compute for that. In addition it is not clear if that computation adjusted for clustering anticipated within each region since from each facility there was more than one facility

4) The ethics statement should be clear as to who gave consent and who gave assent. In addition, clarify if the verbal consent was documented

5) Write IFA appearing on line 149 in full the first time it is appearing

6) In Line 192 on page 8, take care of the typographical error of "concentration"

7) The analysis does not show that clustering was adjusted for. The confidence intervals for the proportions also reflect this omission. This calls for re-analysis to adjust for this clustering

8) The analysis should also indicate if attempts to check for interaction were made and confounding were made. The results does not seem to have eliminated any factors that were included at multivariable analysis. Were they all retained as confounders? What exactly were the non-significant factors that were retained in the final model confounding? Or were they retained because the model building was incomplete?

9) The results under "predictors of infection with malaria" from line 264-276 should be put together. The odds ratios have been put separate from the p-values creating some form of repetition

10) The wide confidence interval of "farmer" category in the occupation should be noted as a limitation likely because of inadequate numbers in categories hence random error

11) It is not clear why the associations with diversity score results were analyzed both in the categorized and numerical form. Is there something different the two results are showing. The conclusions from the two methods seem similar in terms of which factors are associated

12) In addition there are factors with more than two levels but which have only one regression coefficient (betas). It is not clear why this is so. Also, the confidence interval and related p-values are adequate, it is not necessary to add standard errors. Also, even for those variables with two levels it is not clear which one is the reference category

13) The analysis looking at associations between dietary diversity and malaria does not seem to be adjusted for potential confounders

6. PLOS authors have the option to publish the peer review history of their article (what does this mean?). If published, this will include your full peer review and any attached files.

Reviewer #1: **Yes: **CHARLES APPREY

Reviewer #2: **Yes: **Joan N Kalyango

---

## [Author Response · Author response to Decision Letter 0]

31 May 2023

Dear Editor 

Thanks a lot for your email of 15 March 2023 providing the reviewer’s comments on our manuscript and for inviting us to submit a revision. We have now prepared a revised version of the manuscript reflecting the suggestions requested by the Editor and Reviewers for consideration. Below we have provided a point-by-point response to the concerns and comments detailing the changes we have made in response to each point raised. The changes made to the manuscript based on the comments and concerns of the Editor and reviewers have been tracked in the revised manuscript version for appraisal. We would like to thank the reviewers for taking the time to read our manuscript again and to provide thoughtful and helpful comments for revision. We believe the revised version of our manuscript represents a significant improvement on the original.

We look forward to hearing from you in due course 

Vanessa Tita Jugha (on behalf of all authors)

See Authors’ responses to Editor and Reviewer’s comments in the attached document.

Responses to Editor and Reviewers’ Comments and Concerns

Manuscript Title: Association between malaria and undernutrition among pregnant women at presentation for antenatal care in health facilities in the Mount Cameroon region

Authors

Vanessa Tita Jugha1*, Juliana Adjem Anchang2, Germain Sotoing Taiwe1, Helen Kuokuo Kimbi1,3,4, Judith Kuoh Anchang-Kimbi1

V2

Date: 13th May, 2023

Please see authors’ responses to Editor and reviewers’ comments and concerns over leaf.

Response to Comments and Concerns of the Editor

 The corrections in the naming of the files and author affiliations have been effected.

 Data underlying study findings have been uploaded in the supporting information files section as S1 Manuscript Dataset.xlsx 

Reviewers' comments to authors and responses

The changes made in the revised version of our manuscript with respect to the reviewers’ comments and concerns have been highlighted in yellow and blue in the manuscript with tract changes for easy appraisal.

Reviewer #1: REVIEWER’S COMMENT

Abstract

Point 1: Result for Maternal Dietary Diversity Score needs to be reflected in the abstract.

 The result for maternal DD score has been included in the abstract (line 35-37) as requested by the reviewer and is highlighted in yellow.

Point 2: Line 39: the word “conversely” (used for reverse relationship between two statements) is a misfit. “Also” can be used instead.

 The word “conversely” has been changed to “also” (line 41).

Point 3: Line 42: … were identified as risk factors of DD… I suggest you omit “risk factors” and use “predictors” instead.

 The change from “risk factors” to “predictors” has been effected and is highlighted in yellow (line 44).

Point 4: There was no conclusion for the abstract.

 The conclusion (highlighted in yellow) has been included in the abstract section of the manuscript as requested. Line 45-49.

Point 5: keywords were not listed.

 Key words (highlighted in yellow) have been included as requested in the revised version of the manuscript line (51-52).

Introduction

Point 6: Line 76-77: … furthermore, the consequences of inadequate maternal nutrient intake for the newborns are equally serious … Please rephrase as “… furthermore, the consequences of inadequate maternal nutrient intake are severe…”.

 The change requested (highlighted in yellow) have been effected in the revised version of the manuscript (Line 78-79).

Point 7: Line 80 -81: Please explain further why anthropometry is often used in low income settings to measure maternal undernutrition.

 Explanation as to why anthropometry is often used in low income settings to measure maternal undernutrition have been provided (line 81-83) and highlighted in yellow. 

Methods

Point 8: Line 106-108 and 114-116 (the longitudes and latitudes) should be omitted.

 Omission effected as requested. 

Point 9: Line 123-126 can be can be added to the introduction. 

 The reviewer is right however, line 123-125 was purposely included in the description of the study area section of manuscript to give an overview of malaria transmission pattern and prevalence rates in the study population. This is because this section is specific to the Mount Cameroon area which is the study area. 

Point 10: Line 135-137 should form part of the ethics statement.

 The change (highlighted in yellow) requested has been effected. line 145-147.

Point 11: Data sampling technique was not stated.

 Data sampling technique have been stated, line 135-137. highlighted in yellow.

Point 12: Some methodological queries including validation of data collection tools like the questionnaire, the constitution of the team that performed the data collection, training provided to the team were not addressed.

 Methodological queries (line 149-150) have been addressed. It is highlighted in yellow.

Point 13: Why was marital status not included as it would have peradventure been one of the predictors for inadequate dietary diversity score.

 Inclusion effected (highlighted in yellow). Line 151.

Point 14: Since 24-hour dietary recall was used to assess the maternal dietary diversity, elaborate more on how it was obtained (two weekdays and one weekend).

 Details (highlighted in yellow) on the 24-hour recall method used in the survey have been provided. Line 166-168.

Point 15: Please state the food groups for readers to have a fair idea about it.

 Food groups stated (highlighted in yellow). Line 169-172. 

Point 16: Line 148-158: Socio-demographic data should be separated from the clinical assessment. 

Line 170-176 can form part of the socio-demographic data.

 Socio-demographic data have been separated from clinical assessment (line 157-162). With respect to line 170-176 now line 178-184 (in the revised version of the manuscript), the reviewer is right however, the authors thought the indicators for socio-economic status be listed in the sociodemographic section of the manuscript as this was part of the questions asked during the data collection process whereas the method used to determine study respondents household socio-economic status (from the listed indicators) be explained in another section different from socio-demographic data.

Result

Point 17: For table 1, the percentages for both adequate and inadequate dietary diversity for the MMD-W status must be stated. Same applies to the anaemia status, IFA intake, ITN ownership and ITN usage.

 Percentages for adequate and inadequate dietary diversity, anaemia status, IFA intake, ITN ownership and ITN usage have been included (highlighted in yellow) in Table 1.

Discussion

Point 18: It was stated that irrespective of the level of education, pregnant women in the Mount Cameroon are all exposed to almost the same level of malaria transmission but also stated (line 369-370) that malaria parasitaemia varied significantly with maternal education level. Please clarify the statement.

 The statement has been revised (highlighted in yellow) for clarity, line 376-382.

Conclusion

Point19: State whether there was an association between the predictors of malaria infection and inadequate dietary diversity since study was establish a relation between the prevalence and predictors of malaria infection and inadequate

 Of the predictors of infection with malaria parasites (Table 3) the only factor significantly associated with inadequate dietary diversity (DD) was maternal occupation (Table 4). Nevertheless, the principal aim of this study was to determine if an association existed between malaria infection and inadequate DD. This observation was stated in the conclusion (line 465-466). In addition to this, predictors of malaria infection and predictors of inadequate DD were also determined. 

Reviewer #2: 

1) The aim of the study should be synchronized in the abstract, introduction, discussion, as well as with the title of the study. In addition, the undernutrition should be consistently referred to. In the title, it is referred to inadequate dietary diversity score, in the abstract it is referred to as undernutrition, and in the introduction it is inadequate dietary intake. However, reference 24 in this study shows that findings on dietary diversity score were already published. So it is better that as the aim is synchronized, this part is omitted to avoid dual publication

 Study aim have been synchronized in the abstract, introduction, discussion and title section (highlighted in blue) of the revised version of the manuscript.

2) The results on prevalence of undernutrition are missing from the abstract. If this is still part of the aims, include them.

 The result on the prevalence of undernutrition have been included in the abstract (line 35-37).

3) The sample size estimation presented has been shown to be based on prevalence of anemia. However, it should be shown whether the sample size is adequate to answer the objectives of the current study. In addition, the estimation is for descriptive study yet the study has analytical component (the associations). Should compute for that. In addition, it is not clear if that computation adjusted for clustering anticipated within each region since from each facility there was more than one facility.

 The reviewer is right. This study is a follow up of a previously published work and is a sub objective of a major study in the area whose overall outcome was anaemia. Moreover, both malaria and/ poor nutrition can lead to anaemia. Thus, anaemia prevalence was used to compute the sample size because according to the World Health Organization it is still a severe (≥40%) health problem in the study area (World Health Organization. ‎2011. Haemoglobin concentrations for the diagnosis of anaemia and assessment of severity. World Health Organization. https://apps.who.int/iris/handle/10665/85839). This justification has been included (highlighted in blue) in the revised version (line 129-130) of the manuscript.

Furthermore, this cross-sectional study was not only analytical but descriptive as well. After calculating the sample size, it was adjusted for sensitivity analysis by adding a 10% non-response rate, and only women who gave their consent were enrolled into the study using the non-probability convenience sampling technique. With this sampling technique participants were approached (convenience) and those who gave their consent were enrolled consecutively. Each pregnant woman irrespective of trimester of pregnancy was seen/enrolled only once and not followed up. 

4) The ethics statement should be clear as to who gave consent and who gave assent. In addition, clarify if the verbal consent was documented

 The ethics statement has been revised for clarity. Line 142-145.

5) Write IFA appearing on line 149 in full the first time it is appearing

 IFA has been written in full. Line 151.

6) In Line 192 on page 8, take care of the typographical error of "concentration"

 Correction noted and effected Line 199.

7) The analysis does not show that clustering was adjusted for. The confidence intervals for the proportions also reflect this omission. This calls for re-analysis to adjust for this clustering.

 Thank you for your observation. This study employed the use of convenience sampling which is a non-probability sampling technique and not cluster sampling. The sample size for this descriptive and analytical cross-sectional study, after calculation was adjusted by taking into consideration a 10% non-response rate. 

8) The analysis should also indicate if attempts to check for interaction were made and confounding were made. The results does not seem to have eliminated any factors that were included at multivariable analysis. Were they all retained as confounders? What exactly were the non-significant factors that were retained in the final model confounding? Or were they retained because the model building was incomplete?

 The reviewer is right. Attempts to check for confounders/interaction between the variables was done using the Variance inflation Factor (VIF). After performing collinearity diagnostics, all VIFs were less than 1 an indication that collinearity between variables was absent. This has been indicated in the statistical analysis section of the revised manuscript. Line 220-223.

 In addition, prior to the multivariable regression analysis, variables with explanatory plausibility and P < 0.2 were subjected to collinearity diagnostics before inclusion into the regression model. This also has been stated in the statistical analysis section of the revised manuscript. Line 220-223.

9) The results under "predictors of infection with malaria" from line 264-276 should be put together. The odds ratios have been put separate from the p-values creating some form of repetition.

 Thank you, the results of the odd ratios under “predictors of infection with malaria” have been put together to avoid repetition as requested. Line 268-286.

10) The wide confidence interval of "farmer" category in the occupation should be noted as a limitation likely because of inadequate numbers in categories hence random error.

 This has been included (highlighted in blue) in the limitation section of the manuscript. Line 449-451. 

11) It is not clear why the associations with diversity score results were analyzed both in the categorized and numerical form. Is there something different the two results are showing. The conclusions from the two methods seem similar in terms of which factors are associated.

 Table 5 has been separated for clarity into: mean DD scores with respect to maternal socio-demographic and antenatal characteristics (Table 5; line 307-319) and multiple linear regression analysis of factors influencing DD (Table 6; line 321-329). The aim of Table 5 was to compare maternal DD scores with variables with explanatory plausibility (health facility type and education) and P < 0.2 from the bivariate analysis. This analysis was carried out to provide a better understanding of how DD scores varied with age, type of nutritional counselling received in the different health facilities, clinic visits attended. In addition to determining the factors associated with dietary diversity, the multiple linear regression analysis was carried to show principal factors influencing DD scores among pregnant women in the study area. Of the factors, antenatal care visit is the only instance pregnant women come together to benefit from nutritional counselling. Thus, study conclusion was made highlighting the benefits of attending antenatal care visits to improve maternal dietary nutrient intake. Prior to analysis in Table 5 and 6, confounders/interactions were checked using collinearity diagnostics. 

12) In addition there are factors with more than two levels but which have only one regression coefficient (betas). It is not clear why this is so. Also, the confidence interval and related p-values are adequate, it is not necessary to add standard errors. Also, even for those variables with two levels it is not clear which one is the reference category.

 Variable used for multiple linear regression in Table 6 have been re-analyzed and adjusted as requested for clarity. Line 321-329.

13). The analysis looking at associations between dietary diversity and malaria does not seem to be adjusted for potential confounders.

 Thank you for the observation. Prior to analysis confounders were adjusted by taking into consideration collinearity diagnostics.

---

## [Decision Letter · Decision Letter 1]

16 Aug 2023

PONE-D-23-01027R1Association between malaria and undernutrition among pregnant women at presentation for antenatal care in health facilities in the Mount Cameroon regionPLOS ONE

Dear Dr. Jugha,

Thank you for submitting your manuscript to PLOS ONE. After careful consideration, we feel that it has merit but does not fully meet PLOS ONE’s publication criteria as it currently stands. Therefore, we invite you to submit a revised version of the manuscript that addresses the points raised during the review process.

We look forward to receiving your revised manuscript.

Kind regards,

Gifty Dufie Ampofo, M.D., Ph.D

Academic Editor

PLOS ONE

Journal Requirements:

Reviewers' comments:

Reviewer's Responses to Questions

**Comments to the Author**

1. If the authors have adequately addressed your comments raised in a previous round of review and you feel that this manuscript is now acceptable for publication, you may indicate that here to bypass the “Comments to the Author” section, enter your conflict of interest statement in the “Confidential to Editor” section, and submit your "Accept" recommendation.

Reviewer #1: All comments have been addressed

2. Is the manuscript technically sound, and do the data support the conclusions?

Reviewer #1: Yes

3. Has the statistical analysis been performed appropriately and rigorously? 

Reviewer #1: Yes

4. Have the authors made all data underlying the findings in their manuscript fully available?

Reviewer #1: Yes

5. Is the manuscript presented in an intelligible fashion and written in standard English?

Reviewer #1: Yes

6. Review Comments to the Author

Reviewer #1: I think the paper has vastly improved and good to be accepted for publication. The authors have also addressed all the concerns raised in my previous review.

7. PLOS authors have the option to publish the peer review history of their article (what does this mean?). If published, this will include your full peer review and any attached files.

Reviewer #1: **Yes: **CHARLES APPREY

---

## [Author Response · Author response to Decision Letter 1]

20 Sep 2023

Dear Editor 

Many thanks for your email of 16 August, 2023 providing comments on our manuscript and inviting us to submit a second revision. We have now prepared a revised manuscript reflecting the suggestions made by the Editor and reviewers for consideration. Below we have provided a point-by-point response to the concerns and comments detailing the changes we have made in response to each point raised. The changes made to the manuscript based on the comments and concerns of the Editor and reviewers have been tracked in the revised manuscript for appraisal. We would like to thank the Editor and reviewers for taking the time to read our manuscript again and to provide thoughtful and helpful comments for revision. We believe the revised version of our manuscript represents a significant improvement on the original.

We look forward to hearing from you in due course 

Vanessa Tita Jugha (on behalf of all authors)

See Authors’ responses to Editor and Reviewer’s comments in the attached document.

Responses to Editor and Reviewers’ Concerns and Comments

Manuscript Title: Association between malaria and undernutrition among pregnant women at presentation for antenatal care in health facilities in the Mount Cameroon region

Authors

Vanessa Tita Jugha1*, Juliana Adjem Anchang2, Germain Sotoing Taiwe1, Helen Kuokuo Kimbi1,3,4, Judith Kuoh Anchang-Kimbi1

Version: 3

Date: 20th September, 2023

Please see authors’ responses to Editor and reviewers’ concerns and comments over leaf.

Response to Comments and Concerns of the Editor

 The list of references has been reviewed to ensure it is complete and up to date as requested. 

Reviewers' comments to authors and responses

The changes made in the revised version of our manuscript with respect to the reviewers’ comments and concerns have been highlighted in yellow, and blue in the manuscript with track changes for easy appraisal.

Point 1: Line 378 states, "Low maternal education levels may facilitate poor knowledge on the utilization and intake of malaria preventive measures..." This can be rephrased as "Low maternal education levels may contribute to inadequate understanding of malaria preventive measures..."

 The correction (now line 375) have been effected as requested. Please see track changes (highlighted in yellow) for the correction.

Point 2: Considering that the dietary diversity score was utilized to assess maternal dietary intake, it is unnecessary to include lines 166-168, which mention the use of 24-hour dietary recall. Both methods can be employed for dietary assessment, but only one is required.

 As requested, lines 166-168 has been deleted in the revised version of our manuscript.

Point 3: The term "undernutrition" should be used consistently, as it is sometimes referred to as inadequate dietary diversity in one context and insufficient dietary intake in another.

 The phrase ‘inadequate dietary diversity’ and ‘insufficient dietary nutrient intake’ have been edited to “undernutrition” for consistency in our revised version of the manuscript. Please see track changes (highlighted in blue) for the corrections.

---

## [Editor Report · Decision Letter 2]

25 Sep 2023

Association between malaria and undernutrition among pregnant women at presentation for antenatal care in health facilities in the Mount Cameroon region

PONE-D-23-01027R2

Dear Dr. Jugha,

We’re pleased to inform you that your manuscript has been judged scientifically suitable for publication and will be formally accepted for publication once it meets all outstanding technical requirements.

Kind regards,

Gifty Dufie Ampofo, M.D., Ph.D

Academic Editor

PLOS ONE
---

## [Editor Report · Acceptance letter]

3 Oct 2023

PONE-D-23-01027R2 

Association between malaria and undernutrition among pregnant women at presentation for antenatal care in health facilities in the Mount Cameroon region 

Dear Dr. Jugha:

I'm pleased to inform you that your manuscript has been deemed suitable for publication in PLOS ONE. Congratulations! Your manuscript is now with our production department. 

Kind regards, 

on behalf of

Dr. Gifty Dufie Ampofo 

Academic Editor

PLOS ONE